# Gender Marginalization in Sports Participation through Advertising: The Case of Nike

**DOI:** 10.3390/ijerph18157759

**Published:** 2021-07-22

**Authors:** Kirsten Rasmussen, Mikaela J. Dufur, Michael R. Cope, Hayley Pierce

**Affiliations:** Department of Sociology, Brigham Young University, 2008 JFSB, Provo, UT 84602, USA; mikaela_dufur@byu.edu (M.J.D.); michaelrcope@byu.edu (M.R.C.); hayley_pierce@byu.edu (H.P.)

**Keywords:** brand activism, gender stereotypes, sports, qualitative content analysis, TV advertising, Nike

## Abstract

The sport sector functions as a site of health-promotion by encouraging and enabling individuals to invest in their health and giving them tools to do so. This investment is often initiated by, or altered by, role modeling, or seeing other individuals engaging in sport. This could include family or peers but could also include depictions of sport in popular media. Inclusive role-modeling could subsequently encourage more sport participation, thus expanding access to health benefits that arise from sport. However, stereotypical depictions of sports role models could make sports seem like a more exclusive space and discourage participation. We examine a case study of a prominent athletic brand and their advertising to examine the ways they expand or reify stereotypes of gender in sport. Through a qualitative content analysis of 131 commercials released by Nike in the past decade, we explore whether their stated goals of being a socially progressive company extend to genuinely diverse and inclusive portrayals of gender in their commercials. Our results indicate that Nike commercials continue to treat sports as a predominantly and stereotypically masculine realm, therefore marginalizing athletes who are female, who do not fit traditional gender binaries, or who do not display traditionally masculine qualities. We also find that the bulk of athletes portrayed by Nike are those who adhere to gender stereotypes. Despite their purported goal of encouraging individuals to participate in sports, Nike’s promotion of gendered sport behaviors may be having an opposite effect for some consumers by discouraging sports participation for those who do not align with the gendered behavior Nike promotes. The stereotyped role modeling of the sport sector portrayed in a majority of Nike commercials could dissuade already marginalized individuals from participating in the health-promoting behaviors available through sport.

## 1. Introduction

Sports participation is associated with several desirable outcomes, including improved physical and mental health [1,2]. While the decision to participate in sport is multifaceted, the portrayal in popular media of sports as fun and desirable, and of celebrity athletes as role models, could help increase individuals’ desires to play sports [3]. This may be especially salient for attracting children and youth to sport participation and creating lifelong activity patterns [4]. Making sport participation attractive to a broad audience, then, can promote greater and more equitable distribution of both sport participation and the health benefits associated with that participation

While some companies claim as part of their mission statements the goal of increasing sports participation and its commensurate positive outcomes, it is logical these companies are even more devoted to extending their consumer pool—the more people they attract to sport, the more consumers they create who will want to buy their products [5]. Still, even if they may have cynical roots related to profit goals, such efforts from companies associated with sports might be seen as a pro-social good if they encourage more individuals to participate in sport and if, in turn, the health benefits associated with that sport participation accrue to both individuals and to improved public health. 

However, such a proposition is built on an important but unproven underlying assumption: that these companies and the media they produce create an inviting picture of sport, rather than one that makes sport seem more exclusive. While there are a number of ways popular culture and media could make sports seem more exclusive, including portraying a limited range of the ideal body or ignoring accessibility issues, we focus here on how these institutions create and sell messages about gender and sports, including for whom sport is meant and what athletes are meant to look like. If advertising creates a more inclusive picture of how gender operates in sport, we might expect to find that these companies’ corporate actions move beyond profit motives to promoting public health. However, it seems likely that corporations’ greatest interest will be in increasing profits, which may lead them to rely on and reify gendered stereotypes about sports in ways that exclude groups traditionally discouraged from sports participation, such as women or people who do not fit gender binaries or stereotypes.

We focus on a case study of Nike and the advertising media they create surrounding sports and sports participation. Media and sports are both domains that construct and reinforce masculinity and femininity, and Nike resides at the intersection of these areas. Nike is one of the most recognizable and valuable brands in the world, and they are well-known for their influential advertisements [6,7]. Nike has been identified as a company that engages in “brand political activism,” wherein they enter the sociopolitical sphere by taking a non-neutral stance on controversial social issues and using sport to promote social change [8] (p. 388). While this is a growing practice that companies continue to adopt, it is yet unclear how far corporate activism efforts extend and whether companies will espouse progressive ideals that may negatively impact their sales or popularity with consumers. As a result, it is also unclear the degree to which corporations are willing to use brand activism to invite previously marginalized populations into sport at the potential expense of alienating existing customers. Nike’s image as a socially progressive company therefore makes their commercials an interesting site to explore whether there is a limit to a company’s brand activism efforts and, in turn, whether such limitations might reify gendered messages about who is or is not welcome in sport. As gender plays an important role in both media and sports, which Nike resides at the intersection of, an exploration of how Nike constructs gender in its commercials may illuminate what sport behaviors they are promoting and the authenticity of its brand activism. By examining the commercials Nike produces we will explore whether their reputation as a socially progressive company extends to genuinely diverse and inclusive portrayals of gender in their commercials, or if they will instead continue to rely on gender stereotypes to sell their products. The answer to this question will help illuminate the degree to which corporations linked to sports can be a useful mechanism in opening up the health benefits of sport participation to a broader population, versus the degree to which they operate as powerful, if perhaps unintentional, obstacles to expanding sports participation and its commensurate health benefits.

## 2. Literature Review

### 2.1. Gender and Media

Mass media exists as a major agent of socialization in our society, with the power to influence what behaviors and values are seen as socially acceptable, as well as affecting how we perceive and interact with the world around us [9,10]. While new forms of media continue to develop and evolve, television commercials remain the most prominent form of advertising that consumers are exposed to, with some estimates suggesting consumers are exposed to close to 29,000 commercials a year [11,12]. As commercials contribute to and reflect societal norms, understanding how commercials portray gender is key to understanding what gender stereotypes exist and what behaviors and roles are viewed as socially acceptable [10].

Considering that most ads on television are 30 s or shorter, commercials are especially likely to utilize gender stereotypes owing to their ability to quickly convey information to consumers [12]. Stereotypes refer to the shared beliefs about the behaviors and characteristics of a social group, which individuals often internalize, subsequently affecting how they perceive themselves and how they behave [13]. Gender stereotypes specifically refer to the traits and behaviors that are appropriate for male and female individuals in a society, and can be context-specific, such as the existence of gender stereotypes in sport [13].

Although men and women are becoming more equal in the number of appearances in commercials, the nature of their appearances still differs by gender [14]. Commercials reinforce traditional gender roles by depicting individuals in a stereotypical manner, such as presenting women in more decorative roles or in ways that emphasize their visual and aesthetic appeal or portraying men as authoritative and in positions of power [14,15]. Companies use commercials to build interest and awareness for their product and are unlikely to include any content that may hinder this goal, such as depicting gender in an unexpected way that may confuse or distract the audience from the product. As the use of gender stereotypes may make it easier for companies to achieve their advertising objectives, they would likely be disincentivized from presenting gender in a way that deviates from stereotypical expectations. If so, these companies and the media they produce may send messages to the public about how gender is supposed to operate in society and what are appropriate spaces for certain kinds of gender to be practiced in [16].

### 2.2. Gender and Sports

Advertising is not the only space that perpetuates gender stereotypes, as traditional forms of masculinity and femininity continue to be constructed and enforced through sports. The field of sports has historically been considered a masculine realm that women were excluded from and is still identified as one of the most male dominated institutions in the United States; these patterns are even more prominent in some other parts of the world [17,18]. As a result, while women are increasingly enjoying the health benefits of sports participation, gendered barriers that define sports and its benefits as the realm of men are still prominent [19]. The world of sports therefore operates as an institution that teaches hegemonic masculine values to male athletes and reinforces a strict definition of masculinity and heterosexuality [20,21]. Masculine hegemony refers to the “culturally idealized form of masculine character” [22] (p. 83), which includes a focus on competition, physical force, heterosexuality, and the subordination of women [21,23]. This form of masculinity reinforces gender stereotypes, producing a narrow definition of “what it means to be a man,” and what sort of masculine behavior is acceptable [23] (p. 232). Individuals who do not embrace traditional masculine ideals may feel discouraged from participating in athletic activities due to this focus on hegemonic masculinity in sport.

The world of athletics strictly constructs and reinforces hegemonic masculine ideals, indicating that many sports and athletic values are not appropriate for women to engage in [20,21,24]. For example, while the passage of Title IX in the United States and increased female participation in athletics may indicate that access to sports has progressed for women, there is evidence that female athletes and women’s sports remain marginalized and trivialized in comparison to their hegemonic masculine counterparts [20,25]. This includes obvious signals of the value—or lack thereof—of women’s sports and female athletes such as lower pay for more successful female athletes and women’s teams compared to men [26]. This marginalization of female athletes may discourage participation among younger female athletes who do not have role models or examples of professional female athletes to model their behavior or goals on, and as a result might mean that fewer girls and women enjoy the health benefits associated with sports participation. Overall, sports continue to be a male-dominated institution wherein hegemonic masculine ideals are constructed and celebrated, and sports that do not reinforce these ideals, such as sports associated with women, are often undervalued or ignored. These difficulties are likely even more pronounced for athletes who do not fit a defined gender binary, such as transgender athletes [27].

Despite changes in athletic participation, most sports are still characterized by masculine traits, framing them appropriate for only male athletes to participate in [24]. For example, it remains common for sports to encourage masculine ideals such as competition, toughness, domination, and physical strength, leading sports in general often to be viewed as inappropriate for women to fully participate in [28]. As a result, female athletes have been marginalized and pressured into specific sports that emphasize their femininity and that are undervalued compared to more masculine sports, therefore allowing women to participate in sports without contradicting the masculine hegemony.

Athletes who participate in sports viewed as appropriate for their gender are able to conform to societal expectations and are therefore allowed to compete without compromising their gender identity [28]. As these athletes comply with gendered expectations, society perceives them much more positively than they do nontraditional athletes who defy gender stereotypes by competing in a sport seen as inappropriate for their gender [29]. However, the number and nature of such sports are limited, which in turn limits participation opportunities. In addition, since sports have been associated with masculinity for most of their history, female participation in most sports is still generally viewed as unfeminine and deviating from gender stereotypes [30]. In addition to excluding cis women, these linkages between hegemonic masculinity and sport also exclude gay men, lesbians, transgender people, and other athletes who not only fall outside definitions of hegemonic masculinity, but who also fall outside a defined gender binary [27].

Breaking gender norms in sports can lead to judgment of the athlete based not on their athletic competence, but instead on the conflicting nature between their gender and their sport and what this may indicate about them and their sexuality [31,32]. Negative responses to nontraditional athletes may discourage some individuals from competing in sports or athletic activities, for fear that they would be judged or ostracized in similar ways for deviating from accepted norms. If so, these individuals would lose out on the health benefits associated with sports participation. Additionally, athletes that do deviate from gendered stereotypes may engage in compensatory acts that emphasize how they conform to gendered expectations in other ways, which have emotional and physical costs in terms of sport participation. This is especially common for female athletes, who highlight their femininity and heterosexuality as a way of apologizing for participating in a masculine activity [30]. As gendered divisions are seen as natural and the masculine hegemony is seen as the ideal, sport as an institution continues to reproduce and reinforce these gendered trends, indicating that nontraditional athletes who deviate from these stereotypes are likely to be marginalized and ignored by mainstream media.

### 2.3. Gender, Sports, and Media

This disinterest and even aversion to nontraditional athletes becomes more salient when we consider what depictions of athletes are presented in the media. The nexus of sports and media has been termed the “sport-media-commercial-complex” [33] (p. 391), wherein sports are not an isolated organization but instead belong to a larger economic network that utilizes sports to advertise a wide range of products to consumers [25]. While the presence of gender stereotypes is something that exists in media generally, there is evidence that gender bias exists in the sports media commercial complex as well [14,25]. Despite female participation rates in sports increasing over time, the related media coverage has not evolved to match these trends, and in some cases has even declined [25] Studies have found that male athletes and typically masculine sports receive the bulk of media attention, with only a small proportion being dedicated to female athletes and sports, which further normalizes the hierarchy between women’s and men’s sports [25,34]. Media coverage of sports may therefore be inadvertently discouraging female viewers from pursuing athletic participation, owing to the lack of representation of elite female athletes as role models in mainstream media.

Beyond media coverage of sporting events, athletic advertising continues to exhibit this same gender bias as well. For example, the bulk of athlete endorsers are male, and female athletes continue to be underrepresented in both product endorsements and marketing campaigns for their sport [35,36]. Additionally, athletic advertisements that do include female athletes are more likely to feature the most stereotypically attractive and heteronormative athlete rather than the most athletically competent [35], and they are often presented in bland, uninspiring ways [34]. These trends may communicate to viewers that athletes are only successful if they exhibit traditionally gendered qualities, which may inadvertently harm and even alienate consumers who themselves do not align with these standards. Additionally, the lack of representation of athletes who break gendered stereotypes in sports may similarly discourage viewers from participating in certain sports or deviating from stereotypes. These patterns may also indicate that advertisers continue to treat men as the main consumers of sports and sports-related products, providing evidence that the sports media complex will favor more traditionally masculine sports and athletes as a way of appealing to their audience.

One way in which sports media perpetuates gender bias within the institution of athletics is through emphasizing presumed biological differences between the sexes. Through maintaining these ‘natural’ distinctions between male and female athletes the media reinforces the masculine hegemony in which male athletes and masculine sports are privileged over female athletes and feminine sports [37]. This distinction also serves to normalize sport as an institution that is segregated by gender, wherein male athletes exist in an entirely different milieu than female athletes [6]. This separation of male and female athletes becomes increasingly problematic when we consider that masculine athletic culture is highlighted and celebrated in the media, whereas female athletes and sports are excluded from mainstream media attention. Perhaps even more problematic are the increasing number of cases in which these gendered “natural distinctions” are used to exclude athletes with specific physical characteristics that do not fall neatly along presumed gender binaries are excluded from competition [38], and the increased media focus on policing such boundaries. This differentiation of gender in sports media is also something that is reflected in the commerce side of sports as well. Athletic apparel companies typically sell men’s and women’s clothing that are marketed and sold separately. This separation in their products may incentivize companies to reinforce and emphasize a gender dichotomy in their advertisements, as a way to distinguish between their men’s and women’s apparel. Women, nonbinary athletes, transgender athletes, and men who do not adhere to or display markers of hegemonic masculinity may opt out of sports participation, and therefore miss out on the health benefits associated with such participation, in the face of repeated media and consumer messaging that sports spaces are not meant for them.

Furthermore, there is evidence that audiences react differently to nontraditional female athletes than they do to athletes whose gender is congruent with their sport. Athletes whose gender is in harmony with their sport have generally been found to garner a more positive response than their nontraditional counterparts do, although this varies by the gender of the audience. Research done by Jones and Greer [29] found that men’s interest waned when viewing a female athlete who competed in a traditionally masculine sport or who did not conform to gender stereotypes. Companies may therefore be incentivized to embrace gender stereotypes, so that their audience will be more likely to maintain interest in the ad and the product being sold. However, while these findings were true of male viewers, female audiences showed an opposite effect wherein they were more interested in female athletes participating in traditionally masculine sports [29]. Therefore, while portrayals of nontraditional athletes in commercials may cause disinterest in a male audience, female audiences may find these types of commercials to be more appealing and engaging [29,39]. This was found to be true only for depictions of female athletes, however, indicating that nontraditional male athletes may still garner negative reactions from both male and female audiences. While companies may employ gender stereotypes to encourage the audience to focus on their product rather than the gendered implications of their commercial, these findings suggest that the portrayal of certain nontraditional athletes may be able to accomplish this goal without relying on stereotypical depictions. However, most companies appeal to both male and female consumers and would therefore be unlikely to include an image in their commercial that would cause disinterest in half of their viewers. When corporations refuse to create media including such female athletes, they are missing out on opportunities to encourage more female participation in sport.

In addition to drawing attention to a product, commercials also influence consumers’ attitudes and behaviors by providing models for them to learn from. Consumers are especially likely to adopt advertised behaviors when they identify with the characters shown on screen, and when these characters are rewarded for their behavior with success or happiness [40]. Commercials are therefore adept at promoting certain consumer behaviors, while simultaneously discouraging others. For example, a consumer may be disincentivized from engaging in certain activities or behaviors if they do not see individuals like themselves being rewarded for their behavior on screen. Specifically, in regard to gender stereotypes, media consumption has also been found to be tied to an increase in sex-typed perspectives and behaviors, indicating a pattern in which gendered messages in commercials are embraced by consumers [41]. This adoption of gendered perspectives and expectations in the sport media commercial complex is consistent across age groups, although young adults are especially likely to gender stereotype sports [24]. Similarly, these patterns are relatively consistent over time, regardless of changes to legal environments and in participation rates [42]. In this sense, advertisers have no incentive to break gender stereotypes and include more socially progressive content concerning gender in their commercials. Considering that young adults make up an important target demographic for many companies, and that consumers are prone to adopting behaviors they see on TV, reinforcing gender stereotypes in sports may be one way in which advertisements appeal to their audience. At the same time, such patterns may mean that young audiences are particularly vulnerable to sports advertising messages that reify gendered boundaries among sports participation and discourage people from traditionally underserved groups from participating.

### 2.4. Brand Activism

Another way companies have begun to appeal to their consumers is by trying to brand themselves as socially progressive. While most corporations have historically avoided taking a stance on controversial issues, it has become increasingly common for companies to enter the sociopolitical sphere [43]. This trend represents an adoption of “brand political activism,” which is a marketing tactic wherein companies use their brand and cultural power to publicly adopt a non-neutral position on a partisan issue [8] (p. 388). Corporations that take on brand activism identities concerning progressive ideals about sex and gender might produce visions and media that are more inclusive and, as a result, draw more people from historically underrepresented groups into sports.

Although both consumers and employees increasingly expect corporations to do this, the controversial nature of the issues mean that brand activism attempts may often garner negative reactions [44]. Additionally, consumers may view brand activism as simply “woke washing” [43] (p. 445) or an inauthentic marketing ploy in which companies appropriate social justice movements to appeal to consumers without any reflection of these same values in their purpose, ethics, or corporate practices [45,46]. Engaging in brand activism may therefore encourage consumers to scrutinize the disconnect between a company’s advertised message and their corporate practices rather than the intended purpose of building interest in a brand or product [43]. This scrutiny may be detrimental to a company whose brand activism is viewed as disingenuous, as perceived authenticity is found to be an important predictor of marketing success [47].

Although there is a growing body of research that explores brand activism and its presence in marketing campaigns, it is still relatively unclear how genuine and extensive brand activism efforts are. Some companies may be vocal about one issue, such as racial equality, while ignoring others entirely, indicating that there may be certain issues, such as gender equality, that are passed over in favor of other forms of activism. Gender inequality is an important issue that is manifested in many institutions, making the intersection of several of these fields, such as in the sports media commercial complex, an interesting lens through which to determine how far a company’s progressive politics will extend.

### 2.5. Nike as a Case Study

Sports, media, and business are all sites wherein gender is constructed and enacted, making the convergence of these areas important for understanding the social construction of gender in our society. As Nike resides at this intersection of retail commerce and sport, their advertisements provide an interesting site for exploring the construction and presentation of gender in the context of brand activism and in how corporate actions might encourage or discourage certain groups from sports participation. Nike is known for their influential commercials, which emphasize athletic excellence and highlight desirable forms of behavior for athletes [6,48]. Nike is an oft-cited example of a company that engages in brand activism campaigns, and their publicized commitment to progressive ideals and equality is evident through the company’s stated purpose, which includes creating “an equal playing field for all,” and “breaking down barriers for athletes” [43,49]. Such activities could potentially create media that make sports participation seem more likely for people who do not fit traditional ideas of hegemonic masculinity.

However, though Nike has become a visible and vocal proponent of equality and civil rights, some may view these claims as inauthentic. For example, Nike made the controversial decision to endorse noted Black Lives Matter supporter Colin Kaepernick as the spokesperson for their 30th anniversary ‘Just Do It’ campaign, while simultaneously continuing to sponsor the NFL team that had rejected Kaepernick for his protests [43,50]. Further inauthenticity claims may be supported by criticisms of Nike’s culture of gender discrimination and inequality at their headquarters in Oregon [51]. For example, female employees reported being excluded from leadership positions within critical divisions, such as basketball, and marketing campaigns for women’s categories were reported as receiving smaller budgets than traditional men’s sports [51]. Nike as an organization is inherently shaped by the patriarchal world of sports it supports, indicating that the commercials Nike produces and the values they profess may reflect the masculine hegemonic trends seen in the overall athletic sphere as well as their corporate offices [6]. These inconsistencies between Nike’s brand activism efforts and their company practices call into question how progressive Nike actually is, especially in regard to gender equality. Therefore, while Nike may present itself as a feminist company, there is evidence to suggest that this progressive nature might not extend to the inclusion of nontraditional athletes whose gender presentation deviates from stereotypical expectations of gender in their commercials. If this is the case, Nike may, in spite of their claims to brand activism, be replicating gendered messages that discourage sports participation among historically excluded groups.

The messages about gender that commercials convey have been embraced by consumers, indicating that Nike has the power to perpetuate or discourage the adoption of gender stereotypes in society [41]. Despite their purported progressive ideals, Nike likely benefits more from the continuation of these trends than any changes to them. If Nike’s audience adopts the stereotypically gendered trends they see portrayed, they will presumably expect to shop with a brand that clearly distinguishes between men’s and women’s clothing, where they can purchase gendered clothing that allows them to align with the stereotyped representations they have seen. However, if Nike were to consistently and prominently display nontraditional athletes who cross the gendered lines of sports in their commercials, Nike’s approach of clearly delineating their products by gender may seem problematic and be called into question. Therefore, despite their professed commitment to equality, there is evidence to suggest that Nike’s commercials will instead perpetuate gender divisions by only depicting athletes in sports that are seen as appropriate for their gender.

Despite Nike’s success in the athletic apparel market and influence as an acclaimed advertiser, relatively few studies have explored the implicit messages included in Nike commercials. The few that have examined these ads did so prior to 2010, missing much of Nike’s renewed push to brand itself as a socially progressive company. This study will therefore attempt to address this gap in the literature by examining video commercials produced by Nike in the past decade to determine if their brand activism efforts and purported commitment to gender equality is reflected in their commercials. If they are as progressive in regard to gender as they claim to be, we would expect to find equal representation of both male and female athletes, as well as depictions of athletes competing in sports that are nontraditional for their gender and defying gender stereotypes in general. If this is the case, the media Nike produces should be more inviting to women and other people who do not fit hegemonic masculinity stereotypes, which may translate into more sports participation among those groups. Gender equality in the commercials will be determined by examining an athlete’s gender, their gender congruency with their sport, gendered markers, and their presentation overall. This analysis will therefore allow us to determine whether Nike’s brand of inclusion and equality is reflected in progressive representations of gender in their commercials, or if they will instead continue to rely on gender stereotypes to sell their products.

## 3. Methods

### 3.1. Data

To determine the representation of gender in Nike commercials as it relates to their brand activism, we performed a content analysis of commercials released by Nike from 2010 to 2019, inclusive. We chose video commercials over other forms of advertising due to their ability to reach wide audiences through their accessibility on both television and the internet. Additionally, because of our interest in Nike’s efforts to be seen as socially progressive, we limited the commercials in our sample to include only those released in the past decade, as these years capture recent brand activism efforts, such as partnering with Kaepernick in 2018. We accessed these commercials through adforum.com, an online repository of advertisements, which hosts just over 200 video commercials released by Nike in this timeframe. In our sample we included only English-language live-action video commercials that portrayed at least one athlete. We define an athlete as either a professional athlete shown in any activity or an actor shown in an athletic capacity, and a commercial must include at least one of these depictions to be included in the sample. This results in a total sample of 131 advertisements, which constitutes that totality of video commercials released by Nike in this timeframe that met the criteria outlined above.

More than one athlete from each commercial was included in the analysis, resulting in a total sample of 675 athletes. We included an athlete in the analysis if they met a series of criteria. First, they must meet the guidelines outlined above, of either being a professional athlete or an actor shown in an athletic activity. All professional athletes shown in a commercial were included in the sample, and any athlete—professional or otherwise—shown on screen by themselves was also included. If there was more than one athlete on screen then an athlete’s inclusion in the sample was determined based on whether they were in the foreground, had speaking lines, played a central role, or were otherwise distinguished from the other athletes on screen [52]. Nike is unlikely to defy gender stereotypes through an athlete in the background, which is why we only included athletes that played a prominent role in a commercial.

### 3.2. Coding Approach

We adopted a two-stage coding approach (see Table 1) for each commercial. The first stage involved identifying initial codes of interest, which largely consisted of discerning demographic details for each athlete, such as their gender, race, and sport. The second stage involved identifying the broader themes and implicit messages unique to each commercial, which we ascertained by noting the visual images, audio cues, and overall portrayal of each athlete [11,15,53,54].

#### 3.2.1. Initial Codes of Interest

The first code of interest was the gender that each athlete presented. As it is impossible to confirm every athlete’s or actor’s gender identity, we rely on secondary sex characteristics and the presentation of stereotypically masculine or feminine traits to categorize an athlete as either male-presenting or female-presenting (hereafter referred to as male or female) [55]. Gender presentations outside of the male–female dichotomy were only possible to confirm for professional athletes whose personal information was accessible online, all other athletes were categorized based only on their appearance and behavior in the commercial. Examples of stereotyped depictions include female athletes having longer hair or wearing makeup, and male athletes having defined muscles or facial hair [11,17,35].

As race and gender are both constructed through sports, we also identified the race of each athlete. We used a variety of methods to assign a racial category to each athlete, including pictorial evidence, articles written about the commercials, information about what country the commercial was produced in, and online biographies about professional athletes. We ultimately assigned each athlete to one of six categories: Black, White, Latinx, Asian, Multiracial, and Other. The Other category represents individuals who are not White but whose racial identity we are unable to confidently determine. Additionally, we employ an ‘unknown’ category in situations where there is an obstructed view or face covering that inhibits us from identifying their race.

Following the gender and race of each athlete we then distinguished the sport or athletic activity they were portrayed in and assigned them to one of 30 possible categories. The bulk of these categories represent specific sports (basketball, softball), while one is non-specific, for generic athletic activities that are not tied to a specific sport, such as working out. The full list of sports is displayed in Table 2. We then categorized the gender associated with each sport as either traditionally feminine, masculine, or neutral. These classifications are determined following methods used in previous research, as well as high school participation rates by gender from the 2018–2019 school year [28,32,42,56]. Attributes of typically masculine sports include high levels of bodily contact, force, strength, and aggression [28,56]. These sports are viewed as appropriate for male athletes to participate in, and include football, weightlifting, rugby, basketball, skateboarding/BMX, snowboarding, surfing, baseball, boxing, wrestling, lacrosse, cricket, karate, fencing, hockey, and water polo.

Traditionally feminine sports usually do not involve high levels of physical contact or face-to-face opposition and instead are often individual activities that emphasize aesthetically pleasing movements [32,42,57]. These sports are viewed as generally suitable for female athletes, and include gymnastics, volleyball, softball, dance, and ice skating. Though not a sport in the traditional sense, the inclusion of yoga in Nike’s commercials also led to the classification of yoga as a traditionally feminine sport. Neutral sports are viewed as acceptable for either gender to participate in, and include soccer, swimming, track and field/running, cycling, triathlon, tennis, and golf [42]. While some may view these sports as slightly more masculine or feminine, high school participation rates in the U.S. indicate that these sports are popular for both genders [56]. The non-specific category of working out is also considered gender neutral.

Finally, after discerning each athlete’s gender and the gendered nature of their sport, we determined whether the individual’s gender presentation aligned with the gender traditionally associated with their sport. We considered an athlete gender congruent if they were a female athlete competing in a traditionally feminine sport or a male athlete competing in a traditionally masculine sport. Examples of gender congruent athletes include female athletes shown dancing or doing yoga, or male athletes depicted playing football or basketball. We also considered any athlete depicted in a gender-neutral sport as gender congruent, such as a female athlete playing soccer or a male athlete playing tennis. A male athlete participating in a traditionally feminine sport or a female athlete participating in a traditionally masculine sport was considered nontraditional and gender incongruent. Examples of nontraditional athletes include male athletes dancing or ice skating, or a female athlete boxing or playing hockey.

#### 3.2.2. Broader Themes

A variety of other important factors beyond these initial codes were accounted for in each commercial, both regarding the individual athletes as well as the commercial overall, which helped illustrate the implicit messaging and presence of gender stereotypes in each commercial. Details regarding the specific portrayal of each athlete helped us assess these broader themes. How the athlete is dressed, such as wearing baggy or form-fitting clothing, and camera focus on specific body parts, such as their torso or legs instead of their face, can convey a great deal about how an athlete’s gender is constructed. Similarly, how the athlete moves on screen and their level of activity, such as whether they are shown sitting or sprinting, and their exertion level, as indicated by labored breathing or perspiration, can additionally illustrate gendered differences in presentation. Furthermore, how an athlete interacts with other athletes, such as helping a teammate or struggling against an opponent, and their overall attitude and emotion, whether it be aggressive, excited, or frustrated, can also reveal how an athlete’s gender is affecting how they are portrayed.

Factors not specific to a single athlete but to the commercial overall were additionally important in developing the general themes of a commercial. We recorded a transcription of dialogue and narration as well as a description of visual images for this purpose. The auditory qualities of a commercial, including the genre of music employed, such as hip hop or classical, the gender of the narrator (if any), the explicit message spoken by a narrator or actor, and diegetic sound emanating from the athletes and activities on screen, such as an athlete grunting or cheering from a crowd, all work together to create an implicit gendered message. Similarly, visual qualities, including the setting, such as a house or a gym, background characters, such as families or teammates, and the overall environment, including stormy weather or muted colors, also contributed to the gendered themes of each commercial and were therefore included in analysis.

## 4. Results

### 4.1. Sports as Masculine

The results of our analysis indicate that Nike continues to treat sports as a predominantly masculine realm in which male athletes compete. This may help to reify boundaries around sports participation that tell women and others who do not conform to hegemonic masculinity stereotypes that sport is not for them. There was a total of 675 athletes featured in this sample of commercials, of which 65% were male athletes, 35% were female athletes, and 3 athletes (0.45% percent) were outside of the male–female binary. In addition to depicting almost twice as many male athletes as female athletes, close to half (47% percent) of the commercials in our sample did not include any female athletes at all. In comparison, only 21% of commercials featured no male athletes. These proportions begin to illustrate a preference in Nike’s commercials for male athletes, and an analysis of the sports these athletes are shown participating in indicates similar trends. In regard to the representation of different sports, less than 6% of athletes were shown competing in a sport that is considered appropriate only for female athletes, despite 20% of all sports presented by Nike being categorized as traditionally feminine. Additionally, female athletes were more frequently shown in masculine sports (22%) than they were in feminine sports (14%), indicating that Nike underrepresents feminine sports in favor of privileging masculine sports, even among female athletes. These trends in representation of both different athletes and different sports indicate that male athletes and masculine sports continue to hold the preeminent position in Nike’s advertising, and that Nike’s brand activism efforts do not extend to gender equality in their commercials in ways they might profess. Nike’s supposedly progressive brand activism is not working to create a more inclusive arena in which a broader segment of the population might enjoy sports participation and the related health benefits—at least not in terms of gender.

A comparison of race and gender also illustrates interesting trends in Nike’s commercials. The majority (67%) of athletes portrayed by Nike are nonwhite, indicating that Nike may care more about portraying a racially diverse cast of athletes than they do about portraying equality between male and female athletes. For instance, almost all nonwhite racial categories, including Black, Asian, Latinx, and Multiracial, include at least twice as many male athletes as female athletes (see Table 3). Therefore, while Nike seems to embrace racial diversity in its commercials, this is seemingly done at the expense of including female athletes of color. For example, the single largest category of athletes are Black male athletes, who are included in Nike’s commercials more than three times as often as Black female athletes are.

This trend, however, may merely reflect Nike’s investment in the NBA and its players, 74% of which are Black [58]. However, the WNBA is similarly 69% Black, and yet Nike continues to underrepresent female athletes of color, indicating that the disparity in representation is due to Nike’s own preference for male athletes and sports, rather than a shortage of nonwhite female athletes [58]. Furthermore, the differences in representation of athletes of color compared to White athletes indicate interesting trends regarding the intersection of gender and race in sports. For example, though Nike generally underrepresents female athletes in their commercials, the White category is the only racial category in which representation is evenly split between male and female athletes (51% and 49%). This may indicate that there are privileges afforded to White women that Nike does not similarly grant to women of color, such as being viewed as normal athletes and featured in commercials at the same rates as their male counterparts.

### 4.2. Reinforcement of Gender Divisions

The vast majority of athletes that Nike includes in its commercials adhered to gender stereotypes in sports, with 92% of all athletes competing in a sport that aligned with their gender presentation. Nike showed male athletes in traditionally masculine sports 70% of the time and in neutral sports 29% of the time, while they showed female athletes in neutral sports 65% of the time and in masculine sports 22% of the time. Traditionally feminine sports comprise the smallest proportion of athletes, with 14% of female athletes and just 1% of male athletes shown participating in a feminine sport. These trends provide further evidence that Nike privileges male athletes and sports, as the majority of sports portrayed were ones that are acceptable for male athletes to participate in. These commercials reify symbolism that tells women that sports is not a space for them.

Kevin Durant and Serena Williams are both elite athletes who are each featured in nine of Nike’s commercials, and they typify how male and female athletes in gender appropriate sports are portrayed. Commercials that feature Durant, a professional basketball player, are characterized by fast-paced footage of him playing basketball, which frequently show him training hard and dominating his opponents. He is often shown wearing loose fitting clothing, and the commercials emphasize his nature as the ‘baddest’ player. Williams, a professional tennis player, is conversely often depicted in skirts and jewelry, and the themes of her commercials are more often about her identity as a female athlete. Furthermore, her commercials are slower paced, and portray her as physically distanced from her opponents and other athletes. While many of the distinctions between how Durant and Williams are portrayed may be attributed to the differences inherent to their sports, their portrayals illustrate how Nike chooses to highlight athletes competing in sports in which their behavior aligns with gendered expectations.

Additionally, comparisons between male and female athletes within the same sport are difficult given the disparity in their representation. For example, Nike portrays 19 different NBA players in this sample of commercials, while only including four WNBA players. The variations in their representation also illustrate the different approaches Nike takes when portraying male and female athletes in the same sport. As basketball is a traditionally masculine sport, the male basketball players are viewed as normal and are allowed to embrace the masculinity of their sport, leading to the inclusion of footage of players like LeBron James and Kobe Bryant dunking on opponents and making impressive shots. Conversely, female basketball players are seen as nontraditional and there are subsequently only two instances in the entirety of this sample in which a WNBA player is shown actively playing basketball.

One of these instances is in the 2011 commercial Spotlight, which features footage of 12 individuals playing basketball around the United States. One of these athletes is WNBA player Sue Bird, who is shown shooting baskets in a pop-a-shot game in an arcade, while NBA players such as Dirk Nowitzki and Kevin Durant are conversely shown making three-pointers in actual basketball games. Her portrayal in an arcade serves to trivialize both her accomplishments in the WNBA and her skills as a basketball player, especially in comparison to her NBA counterparts. Additionally, because she is not portrayed on the basketball court, Nike is able to portray her with her hair down and wearing non-athletic clothing, which serves to further minimize her identity as a professional basketball player.

The only other example of a WNBA player actively playing basketball is in the 2019 commercial Dream Crazier, which features footage of Lisa Leslie dunking during a 2002 WNBA game. Though this portrayal is certainly more progressive than Sue Bird’s depiction, Leslie is depicted with ribbons in her hair, a common apologetic approach used to emphasize an athlete’s femininity. This commercial also features no fewer than 24 other athletes, indicating that Nike will highlight female athletes in masculine sports when they are able to emphasize the athlete’s femininity and surround them with a variety of other athletes that help mitigate their nontraditional nature. Furthermore, the inclusion of a female basketball player dunking in one commercial is hardly comparable to the assortment of commercials that are devoted to individual NBA players, such as the 2018 commercial Rise. Grind. Shine. Again, which focuses solely on Durant’s dedication and abilities as a basketball player and shows him dunking seven times. Overall, even though Nike does include a handful of female basketball players, the fundamental differences between the portrayals of WNBA and NBA players indicates that Nike depicts male and female athletes differently, both within and across sports.

This commitment to portraying male and female athletes in accordance with hegemonic masculine ideals provides further evidence that Nike’s brand activism is likely an insincere marketing approach and not a genuine reflection of progressive values. Nike chooses to enforce a narrow definition of appropriate athletic behavior, which may in turn discourage athletic participation among those who do not align with gendered stereotypes. Further, Nike’s unwillingness to show elite female athletes actually playing their sports cuts off potential role modeling of women experiencing the physical health benefits of sports participation.

### 4.3. Nontraditional Depictions

While the vast majority of athletes Nike portrayed adhered to gendered divisions in sport, 8% of the athletes are portrayed in a manner that is non-stereotypical. These include athletes who are depicted in a sport that is not congruent with their gender presentation, as well as athletes whose gender presentation does not align with the gender binary. The bulk of these nontraditional athletes are female athletes who are competing in a traditionally masculine sport, such as basketball or boxing. Considering that sports are historically a masculine realm and the majority of sports are considered appropriate for male athletes, it follows that female athletes would therefore be more likely to break a gender stereotype and compete in a sport characterized by masculine traits. Furthermore, female athletes already break one gender stereotype by competing in sports in the first place, indicating that the barrier for them to break a gender stereotype and compete in a sport characterized by the opposite gender may be less than it is for male athletes.

While the existence of these nontraditional female athletes is encouraging, in terms of how they might provide role models encouraging women into a broader range of sports participation, the actual portrayals of these nontraditional female athletes vary from case to case. Some take a compensatory approach, emphasizing their femininity as a way of apologizing for the masculine nature of their sport [30]. Rebeka Koha and Lauren Fisher, for example, are both female weightlifters that Nike features in its commercials (Just Don’t Quit and Snow Day, respectively). They are both fairly inactive in their depictions, and rather than seeing perspiration or labored breathing like we do with other athletes, these female weightlifters are presented with pristine hair and makeup, and also wearing jewelry and clothes that emphasize their femininity. These portrayals therefore call attention to their femininity to offset the masculinity that characterizes their sport. Additionally, again, these female athletes are not depicted doing the activities in their sport that are associated with physical health payoffs. Another example of Nike employing feminine markers comes from a 2018 Just Do It commercial featuring Caster Semenya. Though her role as a middle-distance runner is congruent with her identity as a female athlete, some view Semenya as nontraditional because her naturally high levels of testosterone indicate to some that she is too ‘masculine’ to compete with other female runners [46,59]. Nike’s commercial celebrates Semenya while also using feminine markers to reinforce her identity as a female athlete, such as depicting her as a baby wearing a pink onesie. While some depictions of female apologetic behavior are more blatant than others, their usage implies that there is something incorrect about these athletes that the commercials need to compensate for.

Other nontraditional female athletes, however, are not portrayed using an apologetic approach but instead are depicted in a way that does not curtail the masculine nature of their sport. For example, the 2019 commercial Dream Crazier features several female athletes competing in masculine sports without any emphasis on their femininity. On screen we see snowboarder Chloe Kim landing a double cork 1080, football player Sam Gordon tackling a player, and a young basketball player dribbling two basketballs, and none of them are dressed or presented in a way that highlights their femininity or minimizes the masculinity of their sport. It is interesting to note, however, that this commercial specifically celebrates female athletes who have broken barriers in sport, perhaps indicating that apologetic portrayals can only be avoided in specific contexts. Overall, while the inclusion of nontraditional female athletes may highlight Nike’s commitment to equality in sport, the fact that these athletes are often depicted in a way that tries to bring them in line with gendered expectations indicates that there is a limit to Nike’s brand activism efforts.

While most of the nontraditional depictions involve female athletes, there are five male athletes who Nike shows participating in traditionally feminine sports, though the contexts of these depictions indicate that male athletes are only allowed to participate in feminine sports in particular circumstances. For example, three of these nontraditional male athletes are depicted together as dancers in a commercial for Nike’s 2017 Pride campaign. Although we cannot confirm these male athlete’s sexual identities, Nike is clearly positioning them as gay by including them in an advert that celebrates LGBTQ+ pride, indicating to the audience that these athletes deviate from gendered expectations through their sexual identities. Since gay men are marginalized in the masculine hegemony and subsequently labeled as nontraditional and deviant from the norm, Nike is then able to diverge further from the dominant form of masculinity and portray them as participating in a traditionally feminine sport. This indicates to the audience that it is acceptable for male athletes to compete in a typically feminine sport only when the athlete has already been marginalized in other ways by hegemonic masculinity, which may communicate to viewers that they will also be marginalized and labeled as deviant if they choose to participate in non-stereotypical sports.

Another example of a nontraditional male athlete comes from the 2013 commercial Possibilities, which encourages athletes to push their limits in sports. A male athlete is shown dancing, and the narrator tells him, “if you can move your hips, if you can dance, move your legs, move your feet, move the ball,” while on screen we watch him go from dancing on a beach with friends to playing in a professional match with soccer player Gerard Pique [60]. In this case the athlete’s depiction in a feminine sport serves only as a starting point he is told to move on from, towards the gender-neutral, and therefore more appropriately masculine, sport of soccer. Though labeled appropriate for both male and female athletes to participate in, the sport of soccer embodies more masculine qualities than dance does, making it a more acceptable choice for a male athlete to participate in. Not only is this athlete told to change sports, but he is then rewarded for switching by being shown playing at the professional level and scoring a goal alongside Pique and the Barcelona team. This depiction implies to the audience that athletic success can only come when an athlete is in the right sport—which for male athletes is anything but a feminine sport—and this may indicate to viewers that they need to pursue a gender appropriate sport if they wish to be successful or welcomed in sporting spaces.

The final example of a nontraditional male athlete is in the 2012 commercial Find Your Greatness, in which we see a male athlete playing volleyball. It is interesting to note, however, that the athlete jumps in the air to kick the ball rather than hitting it with his hands, more closely resembling the sport of soccer than volleyball. As established previously, soccer embodies more masculine qualities than traditionally feminine sports such as dance or volleyball do, making it a more appropriate sport for male athletes to participate in. His participation in a feminine sport is therefore more acceptable, as he is choosing to embody more masculine traits as he plays. These five examples illustrate that Nike continues to reinforce gendered expectations and stereotypes for male athletes, with only an extremely limited number of specific contexts in which participating in a feminine sport is allowed. This exclusion of nontraditional male athletes severely limits the role models that nontraditional male viewers have to model their behavior on, which in turn may discourage their participation in sports in general. Furthermore, while some depictions of female athletes do appear to be genuinely progressive, the same cannot be said for male athletes, who have harsher limits imposed on what is an acceptable portrayal of masculinity, evincing a clear limit to Nike’s brand activism.

In addition to these athletes who are shown participating in a sport nontraditional for their gender, there are also three athletes whose gender presentation does not adhere to the gender binary. These athletes include Chris Mosier, a transgender man; Leiomy Maldonado, a transgender woman; Leo Baker, who is nonbinary. It is interesting to note that Nike chooses to emphasize the nontraditional nature of these athletes rather than obscuring them. For example, Chris Mosier is a triathlete who is featured in the 2016 commercial Unlimited Courage, and his identity as a transgender athlete is explicitly discussed throughout the commercial. Additionally, as previously mentioned, Leiomy Maldonado is a dancer who is featured in a 2017 commercial celebrating Pride Month, and though her transgender identity is not explicitly discussed in the same way Mosier’s is, LGBTQ+ identities are celebrated generally in the commercial. Nike additionally uses the dress and appearance of these athletes as a way of confirming their gender identity, such as depicting Mosier with his shirt off or portraying Maldonado with long hair and an exposed midriff. These portrayals reflect a more genuinely progressive approach to gender, given the controversy surrounding transgender athletes, indicating that there may be some authenticity in Nike’s brand activism [61]. In contrast to the general representation of cis women and lesbians in Nike commercials, these kinds of representations may encourage transgender athletes to participate in sport.

The portrayal of Leo Baker, however, is slightly different from that of Mosier and Maldonado. Baker is featured in the 2018 commercial Dream Crazy, and they are shown skateboarding while the voiceover announces, “don’t believe you have to be like anybody, to be somebody,” a fairly subtle indication that there is something unique about them [62]. Baker’s nontraditional identity is not celebrated in the same way Mosier’s and Maldonado’s are, and instead Nike uses traditional gender stereotypes in how it portrays Baker. As skateboarding is a sport viewed as appropriate for male athletes, Nike portrays Baker in loose-fitting clothing and short hair as a way of adhering to the gender stereotypes of what an athlete in a masculine sport should look like. Therefore, while Nike does feature a nonbinary athlete, they utilize gendered clothing and hairstyles as a way of ensuring that they have not strayed too far from hegemonic masculinity, illustrating another limit to Nike’s brand activism.

### 4.4. Featured Nontraditional Athletes

All of these nontraditional athletes, including both those in a sport nontraditional for their gender and those who do not adhere to the male-female binary, are overwhelmingly shown in commercials that use large casts of athletes; 84% of nontraditional athletes are presented in a commercial that has at least 10 athletes, while only 64% of traditional athletes are. This is likely done to minimize the fact that they are defying gender stereotypes, as audiences may react negatively to Nike’s brand if forced to confront female athletes who display masculine qualities, and vice versa. Interspersing these nontraditional athletes among more normative athletes who adhere to gendered expectations likely mitigates their non-stereotypical nature, while still including them. Nike shows all nontraditional athletes in a commercial with at least one other athlete, with two notable exceptions: Rebeka Koha and Chris Mosier are each featured in their own commercial. These commercials therefore provide an interesting opportunity to explore how gendered presentations differ between a nontraditional female and nontraditional male athlete.

Rebeka Koha is a Latvian weightlifter who Nike features in a 2018 commercial as part of their Just Don’t Quit series. Weightlifting is a traditionally masculine sport that is viewed as acceptable only for male athletes, and Nike works to mitigate Koha’s nontraditional classification as a female weightlifter by emphasizing her feminine qualities. Koha is fairly inactive in her own commercial; with light music playing in the background she spends more time walking and stretching than actually lifting weights. Additionally, her pristine hair and makeup serve to remind the audience she is a female athlete, and the narration’s emphasis of how she feels at home and comfortable at the gym serves to invoke traditional feminine qualities surrounding the home. These feminine markers become even starker when we compare Koha’s commercial to another 2018 Just Don’t Quit commercial featuring Latvian boxer Zaurs Dzavadovs. Dzavadovs’ commercial is much more high energy, featuring upbeat music, quick cuts, and images of him sparring and pushing himself to his limit, as the narrator explains how kickboxing is a serious sport. These two commercials, although both featuring a Latvian athlete in a masculine sport, stand in direct contrast to one another, with Dzavadovs’ highlighting the masculinity of his sport while Koha’s actively works to hide it. Despite a nontraditional female athlete being featured in her own commercial, Nike strives to compensate for Koha’s nontraditional identity as a female weightlifter by emphasizing how she conforms to gender stereotypes in other ways.

Chris Mosier’s commercial, Unlimited Courage, tells a different story about nontraditional athletes in Nike’s advertising. As Mosier is competing in a gender-neutral sport and Nike is able to sort Mosier into the male athlete group, Nike can invoke the privileges associated with normative male athletes and confer them onto Mosier, despite his transgender identity. These privileges include being featured in a commercial that celebrates his nontraditional nature and is solely focused on him and his abilities. The commercial opens with the narrator announcing that Mosier “is the first transgender athlete to make the men’s national team,” immediately indicating that they are in no way trying to diminish his nontraditional nature [63]. Even the commercial’s title of Unlimited Courage indicates that there is something brave about Mosier’s identity as an athlete that should be celebrated. The portrayal of Mosier is quite different compared to the depiction of Koha, providing further evidence that Nike privileges male athletes and sports in its commercials, even among nontraditional athletes.

The portrayal of Leiomy Maldonado, the only other transgender athlete included in a Nike commercial, also provides an interesting contrast to Mosier’s commercial. Though the commercials were produced for different campaigns, making direct comparisons impossible, there are notable differences that may provide further support for the notion that Nike privileges male athletes over female athletes, even among those viewed as nontraditional. Most obvious is that Maldonado’s commercial includes six other athletes, while Mosier’s focuses solely on him. Mosier also speaks throughout his commercial while Maldonado has no spoken lines. Furthermore, Mosier’s transgender identity is explicitly acknowledged and discussed while Maldonado’s is only obliquely referred to through rainbow text and the word ‘equality.’ These differences may be due to the fact that Maldonado is a female athlete shown participating in a traditionally feminine sport, which, as established previously, are two groups that Nike underrepresents in its commercials. Therefore, despite both athletes identifying as transgender, the fact that Mosier is nontraditional in a masculine way appears to grant him certain privileges in his portrayal that are not similarly afforded to Maldonado. While the inclusion of transgender athletes does indicate progress in Nike’s commercials, the differences between male and female athletes—both among traditional and nontraditional athletes—illustrates that Nike values conforming to gender expectations more so than it does engaging in authentic brand activism and being truly progressive in regard to the portrayal of gender in its commercials.

## 5. Discussion

The sport sector can act as a site of health-promotion by enabling individuals to invest in their health and giving them the space, tools, and skills to do so. In order for individuals to see the sport sector as a place where they belong, inclusive role modeling needs to take place, where individuals can see that sport is a place for them and their preferences and attributes. The results of our analysis reveal that Nike commercials do not paint a picture that sport is for everyone. This stereotypical depiction of the sport sector could make sport seem like a more exclusive space and discourage participation, and deny the health benefits of sport, to already vulnerable groups. Nike commercials continue to employ gender stereotypes as a way of adhering to hegemonic masculinity, as evidenced through a preference for male athletes and the marginalization of feminine sports. These findings reflect patterns found in previous studies, that media coverage underrepresents female athletes, and that sport remains an institution that reproduces traditional forms of masculinity [17,20,21,25,33]. Nike’s depictions of athletes in these commercials also illustrate similar findings found in gender and media studies in general, wherein commercials often present women in visually appealing ways, while showing their male counterparts as powerful or strong [14,15]. Portraying athletes in different ways based on their gender reinforces a masculine hegemony that requires the subordination of women to men and separating male and female athletes promotes traditional gender stereotypes about what behavior is appropriate for men and women. Additionally, the vast majority of athletes Nike portrays adhere to gendered expectations in sport, indicating that Nike is unwilling to deviate too much from stereotypical depictions in its commercials.

Nike purportedly believes that sports have the power “to bring out the best in people,” though their reinforcement of hegemonic masculinity clearly communicates that Nike is only interested in bringing out certain behaviors and values in athletes, or even that Nike is only interested in certain athletes [64]. As Nike highlights only specific forms of athletic behavior, their commercials may unintentionally be discouraging some viewers from participating in athletic activities, and, therefore, from reaping the health benefits of sports participation. Viewers who do not perceive Nike as celebrating people who look or act like them may feel excluded from the institution of sports and be unwilling to participate in athletic activities given the potential for ostracization or judgement. Nike’s commercials may therefore be having a damaging effect on its audiences and even promoting negative associations with athletic participation among its viewers.

Additionally, it appears that Nike’s brand activism may be an inauthentic marketing ploy used to appeal to their consumers. Nike’s reliance on gender stereotypes and adherence to traditional gender expectations indicate that their purported commitment to “fostering an inclusive culture” and “breaking down barriers for athletes” does not extend to how they present gender in their commercials [49]. Instead of breaking down gender barriers in their portrayal of athletes, Nike instead chooses to perpetuate stereotypes through reinforcing differences between male and female athletes. Furthermore, Nike’s presentation of a racially diverse cast of athletes may indicate that they prioritize a focus on racial equality in their commercials rather than gender equality. Including a majority of nonwhite athletes in their commercials may be enough for Nike to feel it can label itself as progressive and inclusive, even if its activism seems to be limited to only racial equality. Their efforts to be seen as a feminist company therefore comes across as disingenuous when we consider that they choose to highlight and celebrate predominantly male athletes in their commercials, even in contexts that celebrate racial diversity. These commercials demonstrate that Nike is largely unwilling to take a progressive approach towards gender in their advertising, likely to avoid engaging in a controversy that could negatively impact their sales.

In regard to the nontraditional depictions in Nike commercials, the bulk are of female athletes which indicates that Nike continues to adhere strictly to a masculine ideal that is largely impossible for male athletes to deviate from. Nike’s presentation of a small proportion of nontraditional athletes does provide some evidence that they made efforts to be more inclusive in their commercials, and the idea that Nike might produce media that would encourage nontraditional athletes, especially those outside of the gender binary, to participate in sports is encouraging. While a small number of female athletes are allowed to deviate from stereotypes and compete in traditionally masculine sports, the reverse is not true for male athletes, who appear to adhere to an even stricter definition of what behaviors and sports are viewed as appropriate for their gender. Furthermore, while Nike does include a small proportion of nontraditional female athletes, their use of apologetic approaches that attempt to compensate for deviations from hegemonic norms indicate a clear limit to their progressive politics. Downplaying the athleticism of nontraditional female athletes and instead emphasizing their visual appeal and femininity ensures that these athletes adhere to gendered expectations in other ways, illustrating that Nike will diverge from the masculine hegemony only when this deviation is compensated for in other ways. Furthermore, the finding that Nike frequently portrays these nontraditional athletes in commercials with many other athletes illustrates that Nike will actively work to mitigate the abnormal nature of these athletes whenever it does include them. The limit to Nike’s brand activism efforts is clear; they will, for the most part, only celebrate diverse and nontraditional athletes when the athlete’s deviance can be curtailed or obscured in some way. The repercussions of these corporate decisions are clear messages to women and others who do not conform to hegemonic masculinity stereotypes—unless they do the hard work of “apologizing” for their presence in a traditionally masculine world, sports are not for them.

Ultimately, Nike’s limited portrayal of gender, under the guise of brand activism, may be helping to recreate and naturalize the continuing gender inequalities that exist in work and family life [34,65,66]. The resistance to loosen its grip on hegemonic masculinity with the simultaneous strategic expansion of female portrayals in sport (strategic in the sense that women have more opportunities as long as they look and act as the female stereotype suggests) echoes the “soft essentialism” that Messner [66] outlines. Where the sport sector acts as a site that appropriates the liberal feminist language of “choice” for girls, but not for boys. Additionally, the limited, non-action-oriented presentation of women in sport continues to reify and normalize the hierarchy of men’s and women’s sports while simultaneously avoiding charges of sexism and stagnation in the fight for gender equality [34]. The sport sector has the opportunity to change lives through the initiation of lifelong health habits. It also has the opportunity to change lives by role-modeling positive social change. The idea that sport, and its health benefits, are for everyone would indeed be evidence of social change. However, Nike’s reluctance to fully engage in the efforts that they claim to support, is further harming vulnerable groups by solidifying and perhaps enabling the stereotypes that harm these groups.

Despite this evidence that Nike’s brand activism may be insincere, their authentic celebration of a few nontraditional athletes does indicate progress towards a more genuine brand activism. For example, the fact that Nike produced commercials that highlight Caster Semenya and Chris Mosier does illustrate that Nike is both capable of and willing to take a genuinely progressive stance on gender issues. Some consumers may view aspects of these athletes’ identities as controversial; however, this did not prevent Nike from dedicating whole commercials to these athletes in ways that explicitly support them and do not make apologies for who they are. The issue with Nike’s approach to brand activism, however, is that the support of these athletes is not often backed up in their other commercials. The inclusion of a handful of nontraditional athletes does not outweigh the fact that the overwhelming majority of athletes Nike presents are those that adhere to gender stereotypes. Nike grants more exposure and media attention to these athletes viewed as ‘normal,’ indicating that Nike cares more about aligning with hegemonic masculinity than they do breaking down barriers for nontraditional athletes and providing opportunities and role models in sports for a broader pool of potential athletes.

## 6. Limitations

One limitation of our sample was the high variability in the number of commercials produced each year. For example, there were 21 commercials produced in 2015 that met our criteria, but only five produced in 2019, making comparisons across time difficult despite the breadth in years of the sample. It may therefore be interesting for future research to look longitudinally at how Nike’s marketing approach may have changed over the course of the company’s history, potentially illuminating progress towards genuine brand activism over time. Another limitation comes through the fluidity of gender identities. An individual’s gender may change over time and our reliance on pictorial evidence to determine an individual’s gender may mean that some athletes were incorrectly classified. However, seeing as our research was focused on how Nike depicts gender in its commercials, we are confident that utilizing an athlete’s gender presentation accurately reflects Nike’s stance and representation of gender.

## 7. Conclusions

Overall, Nike’s purported commitment to gender equality and inclusion is not reflected in their commercials. As such, their activism regarding gender equality may come across as insincere. Some viewers may therefore feel alienated or excluded from the institution of sports owing to Nike’s commitment to portraying a narrow definition of appropriate athletic behavior. Consumers may additionally be skeptical of Nike and other brands that take similar activist stances, considering the evidence that these progressive attitudes may merely be disingenuous marketing. Hegemonic masculine ideals continue to be strictly reinforced in both sports and media, and our results provide evidence that even companies who engage in brand activism are restrained by this narrow definition of masculinity and reproduce it in their commercials. Furthermore, in light of Nike’s prominent role in the world of sports media, their commercials may indicate that inclusion and acceptance of nontraditional athletes in mainstream media, and perhaps sports in general, has not been achieved in ways that we as consumers may be led to believe. As a result, the health benefits of sport participation may not be as broadly accessible as they could be. As brand activism becomes an increasingly common practice, further research is needed to explore how the adoption of this approach may be affecting sales and brand image, as well as how consumers are interpreting and reacting to companies who engage in this approach. If such brand activism could be employed to open up sport participation, and the associated health benefits, to a wider audience, a broader discussion of the moral components of corporate-related sports activities is needed.

## Figures and Tables

**Table 1 ijerph-18-07759-t001:** Coding Approach.

Stage 1		Stage 2	
Functions:	Identifying initial codes of interest	Functions:	Identifying the broader themes and implicit messages
Example:	Gender	Example:	Presence of gender stereotypes

**Table 2 ijerph-18-07759-t002:** Percentage of athletes presented in each sport in Nike commercials, 2010–2019.

	Male	Female		Male	Female
Masculine Sports			Neutral Sports		
Baseball	3.0	0.0	Cycling	0.7	1.3
Basketball	34.9	6.4	Golf	4.8	1.7
Boxing	3.0	4.7	Working out	1.8	7.3
Cricket	8.0	0.0	Swimming	0.7	1.3
Football	9.3	1.7	Soccer	13.9	18.9
Hockey	2.1	0.9	Tennis	1.1	7.3
Lacrosse	0.9	1.3	Track/Running	6.2	26.2
Skateboarding/BMX	4.1	1.3	Triathlete	0.2	0.9
Weightlifting	0.2	2.2			
Wrestling	1.1	0.4	Feminine Sports		
Snowboarding	1.1	0.4	Dance	0.9	3.0
Rugby	0.7	0.0	Gymnastics	0.0	3.9
Fencing	0.0	0.9	Ice skating	0.0	0.9
Water polo	0.0	0.9	Volleyball	0.2	2.6
Surfing	1.1	0.0	Softball	0.0	0.9
Karate	0.0	0.4	Yoga	0.0	2.6

Note: male *n* = 439; female *n* = 233; two transgender athletes and one nonbinary athlete are not presented in the table.

**Table 3 ijerph-18-07759-t003:** Demographics of athletes included in Nike commercials, 2010–2019.

	Male		Female	
	Number	Percent	Number	Percent
Gender of Sport				
Masculine	305	69.5	50	21.5
Feminine	5	1.1	32	13.7
Neutral	129	29.4	151	64.8
Nontraditional				
Traditional	434	98.9	184	79.0
Nontraditional	5	1.1	49	21.0
Race				
Black	187	42.6	57	24.5
Latinx	22	5.0	8	3.4
Asian	82	18.7	38	16.3
White	110	25.1	107	45.9
Multiracial	21	4.8	5	2.2
Other	14	3.2	16	6.9
Famous Athlete				
Yes	217	49.4	95	40.8
No	222	50.6	138	59.2

Note: male *n* = 439; female *n* = 233; two transgender athletes and one nonbinary athlete are not presented in the table. There are three male athletes and two female athletes whose race we could not identify.

## Data Availability

Publicly available commercials were analyzed in this study. These commercials can be found here: www.adforum.com.

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
