# Peer review of "Gender Marginalization in Sports Participation through Advertising: The Case of Nike"

_ijerph, 2021, doi:10.3390/ijerph18157759_

Round 1

Reviewer 1 Report

Dear authors,

I suggest rewriting the abstract section, improved the literature review and support recommendations with deeper analyses. The paper is not in correspondence with the journal scope. The paper is exciting but on the average level. Some aspects need to be improved.

Abstract: The authors have to specify what instrumental tools and what concrete research methods have been applied.

Literature review: please add more actual sources. The reference list must be updated considering the most recent works in the field, including those indexed in Scopus and Web of Science databases during the last five years.

Materials and methods: It is necessary to explain why the number of 131 commercial releases is sufficient and relevant for the study. The scientific research soundness will benefit if the authors put the table and use the space to state clearly how and with what dataset the investigation was done and why. Please be more concreate, what methods and methodology have been applied, and why.

Conclusions and policy recommendations: The discussion part is underdeveloped; the comparisons of the results obtained with other similar studies are missing. Please, mention the other research with similar results and extend the discussions part of this paper.

Good luck!

Reviewer 2 Report

I strongly support the work of this study and tip my hat off for the authors to conduct such an important and critical work for gender equity. This is an important work and should definitely be published.

However, when I first read the title and the first half of the abstract, I wanted to reject it because it reads like a corporate promotion for Nike’s gender activisms works, which are all FAKE and ill-intentioned in my opinion.

Thus, the title should be revised to remove the misleading activism, while changing it to more direct representation of the study’s argument, such as “Gender marginalization through branding commercials: the case of Nike”.

Abstract line 12: “Nike engages in brand activism, a marketing approach that uses sport to promote social change.” – please remove. This is a side-story of the paper. The main finding is how Nike is marginalizing females through commercials. This could mislead readers into positive images for evil corporates.

Line 47: “and they are well known for their influential and inspiring advertisements” – inspiring is not a good description of what it is. Remove.

Section 2.4 Brand Activism – the paper is already quite long and this section can be shortened. The study material of the paper focus more on commercials.

 3.2. Coding Approach – this section can be strengthened and clarified with a table.

Line 619: “This provides further evidence that Nike’s brand activism is likely an 620 insincere marketing approach and not a genuine reflection of progressive values” – the findings here are fantastic. The authors should make a separate paragraph to highlight the importance of these findings. Do not mesh them with cases.

Line 863: “However, 865 the finding that the bulk of these nontraditional depictions are of female athletes indicates 866 that Nike continues to adhere strictly to a masculine ideal that it is largely impossible for 867 male athletes to deviate from.” – this sentence should lead and start the paragraph as it is very powerful. 

Line 916: “While certain aspects of their brand activism, such as being 917 a proponent of racial equality, may be authentic and reflect genuine values as a company” – this is of great doubt and not studied in the paper. I suggest not discussing it.

Round 2

Reviewer 1 Report

Dear authors,

 thank you for the changes. It helps. The structure of the article meets the standards. on the other hand, the scientific value of the paper is on the average level. The objectives and aims of the paper are in correspondence with the applied methodology. The scope and recommendations of the article are acceptable.

The article is acceptable for publication.